# Transcriptomic analysis of benznidazole-resistant *Trypanosoma cruzi* clone reveals nitroreductase I-independent resistance mechanisms

**Ana María Mejía-Jaramillo**[ORCID]*[◉], **Hader Ospina-Zapata**[◉], **Geysson Javier Fernandez**[ORCID], **Omar Triana-Chávez**[ORCID]

Grupo Biología y Control de Enfermedades Infecciosas, BCEI, Universidad de Antioquia, UdeA, Medellín, Colombia, South America

◉ These authors contributed equally to this work.

* maria.mejia3@udea.edu.co

## Abstract

The enzyme nitroreductase I (NTRI) has been implicated as the primary gene responsible for resistance to benznidazole (Bz) and nifurtimox in *Trypanosoma cruzi*. However, Bz-resistant *T. cruzi* field isolates carrying the wild-type NTR-I enzyme suggest that additional mechanisms independent of this enzyme may contribute to the resistance phenotype. To investigate these alternative mechanisms, in this paper, we pressured a *Trypanosoma cruzi* clone with a high Bz concentration over several generations to select Bz-resistant clones. Surprisingly, we found a highly drug-resistant clone carrying a wild-type NTRI. However, the knockout of this gene using CRISPR-Cas9 in the sensitive clone showed that NTRI indeed induces resistance to Bz and supports the idea that the resistant one exhibits mechanisms other than NTRI. To explore these new mechanisms, we performed an RNA-seq analysis, which revealed genes involved in metabolic pathways related to oxidative stress, energy metabolism, membrane transporters, DNA repair, and protein synthesis. Our results support the idea that resistance to benznidazole is a multigenic trait. A Deeper understanding of these genes is essential for developing new drugs to treat Chagas disease.

## Introduction

Drug resistance in the parasites that cause neglected diseases is a growing and worrying problem that compromises the effectiveness of essential treatments. Chagas disease or American trypanosomiasis, caused by *Trypanosoma cruzi,* is treated with two drugs, benznidazole (Bz) and nifurtimox (Nfx). Both drugs have unsatisfactory results in chemotherapy, being effective in the acute phase but with a cure rate of only 60–85% [1,2]. Unfortunately, these drugs have several drawbacks, such as treatment failure due to resistance, limited efficacy in the chronic stage of the disease, lack of consistency of treatment response in different geographical areas, side effects that cause 15–20% of people to discontinue treatment, and finally the need for prolonged treatment to increase efficacy [3–9].

**Data availability statement:** ARNseq data are available at: https://dataview.ncbi.nlm.nih.gov/object/PRJNA1182289?reviewer=1cf3ui6nea649ma1usp1ifskp0 All other relevant data are within the paper and its Supporting Information files.

**Funding:** This work was supported by Universidad de Antioquia UdeA, SGR Grant BPIN 2020000100479. There was no additional external funding received for this study.

**Competing interests:** The authors have declared that no competing interests exist.

Bz and Nfx are both prodrugs activated by parasite nitroreductases, resulting in the production of their active metabolites essential for their therapeutic effects. In resistance to Bz and Nfx, the gene encoding the enzyme nitroreductase I (NTRI) has been implicated as the primary response for resistance *in vitro* [10,11] and in patients with treatment failure [12]. Loss of function of this enzyme, due to the deletion of copies and/or mutations within the coding region, reduces its expression and, consequently, its activity, leading to the non-activation of prodrugs and the appearance of therapeutic failure [10–12].

In addition to NTRI, overexpression of genes responsible for detoxifying drug secondary metabolites and underexpression of genes related to drug activation have been implicated as key players in the expression of resistance [13–17]. However, there are still many gaps in our understanding of this phenomenon. Remarkably, Bz-resistant field *T. cruzi* parasites, but with the wild-type NTR-I enzyme, suggest that other mechanisms independent of this enzyme may be involved in the resistance phenotype [10].

To explore the mechanisms underlying Bz resistance, it has been reported that increasing Bz pressures generate resistant parasites in which the NTRI gene is absent or mutated [10,18]. We hypothesized that direct exposure to a high drug concentration for several generations could select parasites with additional genes involved in the drug resistance phenomenon. Indeed, RNA-Seq analysis revealed that the NTRI gene was not affected in the resistant parasites. However, other genes involved in metabolic pathways related to oxidative stress, drug transport, and protein synthesis were affected. The comparison of the transcriptomes of diverse resistant clones with different resistance mechanisms allowed us to find common pathways involved in the phenomenon of Bz resistance, which may represent new targets for the design of future drugs or as possible resistance markers to be further studied in future research. Finally, the results support the idea that the *T. cruzi* parasite has multigenic resistance mechanisms to Bz.

## Materials and methods

### Parasites

*Trypanosoma cruzi* parasites were maintained as epimastigotes at 28 ºC in liver infusion tryptose (LIT) medium supplemented with 10% (vol/vol) heat-inactivated Fetal Bovine Serum (FBS). Clones were obtained by limiting dilution from the WT strain (M-RATTUS/CO/91/GAL-61.SUC; DTU TcI-d; Bz $IC_{50}$ 5.85 µM) and the resistant one. In each case, one clone was selected to perform all the experiments (S1 Fig).

### Determination of the median inhibitory concentration ($IC_{50}$) for benznidazole and nifurtimox

Exponential phase *T. cruzi* epimastigotes were prepared at a final concentration of $5 \times 10^5$ parasites/mL in LIT medium with 10% FBS. In each assay, 10 concentrations of the drugs (0.78–400 µM for Bz or 0.195-100 µM for Nfx) purified from tablets and prepared in DMSO, were evaluated, and each concentration was assessed four times in 96-well plates. Untreated parasites were used as growth controls. Plates were incubated for 72 h at 28 °C, and the effect of the compounds was determined by adding 20 µl/well of alamarBlue (Resazurin sodium salt, catalog number R7017, Sigma Aldrich, St. Louis, USA), prepared in PBS at 0.125 mg/mL and incubating overnight at 28 °C, as was described previously [19]. The absorbances were read at 570 nm and 600 nm in a Thermo Multiskan Spectrum Microplate Reader (serial number 1500-153), and the $IC_{50}$ was calculated using GraphPad Prism 8.0. One-way ANOVA and Dunnett's or Tukey's multiple comparison tests were used to determine differences between $IC_{50}$. Values are expressed as mean ± SD. Statistical significance was determined at $p < 0.05$.

## Benznidazole pressure

To generate benznidazole resistance from the 61S susceptible clone (Bz $IC_{50}$ 6.22 μM), epimastigotes were seeded at 40 μM of Bz purified from tablets (approximately 6x the $IC_{50}$) and subcultured every week under selective pressure for around 20 passages until a resistant population (61R) was established. Then, the 61R was cloned by limiting dilution, and 14 clones were isolated. Finally, the $IC_{50}$ was determined by alamarBlue for the resistant population an its clones as described above.

## Proliferation curves and cell cycle

$1 \times 10^6$ epimastigotes/mL from 61S and 61R_cl4 phenotypes were seeded by triplicate and counted every 24 h for ten days in the Neubauer chamber, and the generation number and doubling time were calculated as described by Mejía-Jaramillo et al. (2009) [20]. The data were visualized and analyzed in GraphPad Prism 8.0. A two-way ANOVA was performed to determine the proliferation, and Sidak's multiple comparisons test was used. Values are presented as mean ± SD. Statistical significance was determined when $p < 0.05$. Additionally, the same number of parasites were synchronized with 10 mM Hydroxyurea (Catalog number H8627, Sigma Aldrich, St. Louis, USA) for 16 hours, labeled with a propidium iodide (PI) solution (Catalog number P3566, Invitrogen, Waltham, USA) and treated with 10 μg/mL RNAase (Catalog number EN0531, Thermo Fisher Scientific, MA, USA). The readings to determine the cell cycle of each cell line were performed on a BD LSRFortessa Cell Analyzer flow cytometer (BD Headquarters, New York, USA), and data were analyzed using FlowJo 10.8 software.

## Transfections

*Trypanosoma cruzi* epimastigotes were transfected by electroporation using the Amaxa® Cell Line Nucleofector® Kit T from Lonza (Catalog number VCA-1002, Lonza, Basel, SW) with the X-001 program in the Amaxa nucleofector II b (Lonza, Basel, SW). For this, $5 \times 10^7$ log-phase parasites were collected by centrifugation and resuspended in 100 μL of the kit solutions with the appropriate nucleic acid (plasmids, sgRNA, and DNA repair templates). After transfections, the parasites were allowed to recover for 24 hours, and the drug selection was performed in 10 mL of the medium. Then, 100 μL of the parasite suspensions were aliquoted in 96-well plates and incubated at 28 ºC. The transfected parasites were obtained approximately three weeks later. Two different transfections were performed: the first to overexpress the Cas9 enzyme in the 61S clone using the pTREX/n-Cas9 plasmid [21], selecting parasites with G418 disulfate salt (Catalog number G8168, Sigma Aldrich, St. Louis, USA), and following the conditions described by Lander et al. (2015) [21]. The second transfection was performed with parasites expressing the Cas9 enzyme (61S_Cas9), using specific sgRNA and a homology repair template (HRT) containing the puromycin gene as described below. These parasites were selected with 250 μg/mL G418, and 60 μg/mL puromycin dihydrochloride from *Streptomyces alboniger* (Catalog number P9620, Sigma Aldrich, St. Louis, USA) according to the protocols described by Queffeulou et al., 2024 [22].

## Western blots

To verify the NTRI expression in susceptible and resistant parasites, protein extracts from 61S and 61R_cl4 epimastigotes in the exponential growth phase were obtained by sonication with extraction buffer (Tris-HCl 20 mM pH 7.9, NaCl 100 mM, sucrose 0.25 M, EDTA 1mM, EDTA 2mM, Triton x-100 (v/v) 0.1% and cOmplete™ Mini EDTA-free Protease Inhibitor Cocktail (Catalog number 11836170001, Roche, Mannheim, Germany)). One hundred (100) μg of proteins were separated by vertical SDS-PAGE and transferred to a nitrocellulose

membrane. The membranes were blocked with 3% milk and incubated with primary antibody nitroreductase (NTRI; 1:500-mouse). The IRDye 800 CW goat anti-mouse IgG1 secondary antibody (Catalog number 926-32350, LI-COR Bioscience, Nebraska, USA) was used at 1:15,000, and blots were imaged using the Odyssey Classic Infrared System (Lincoln, USA) in both 700-nm and 800-nm channels. Since the same amount of the protein from the 61S and 61R_cl4 clones was loaded on the gels, the expression was compared by quantifying the intensity of bands in Odyssey Classic Infrared System. The experiments were performed in triplicate.

### NTRI knockout by CRISPR-Cas9

Double (DKO_NTRI) or single knockout (SKO_NTRI) lines were generated for the NTRI gene (C4B63_56g60) using transfected parasites with the pTREX/n-Cas9 plasmid (61S_Cas9) [21]. Two targeting CRISPR RNAs (crRNAs) were designed using the http://grna.ctegd.uga.edu/ tool, synthesized by the CRISPRevolution service (Synthego, California, USA), and used according to the manufacturer's instructions. The homology repair template (HRT) was amplified from a plasmid containing the puromycin gene using two oligos, with either 30 nucleotides of the 5'-UTR or 3'-UTR from the NTRI plus 20 nucleotides of the Pyrimidine-rich sequence (Y) and puromycin, respectively. Briefly, each sgRNA was resuspended in nuclease-free water, and 4 μL of each and 6 μg of the repair template were used to transfect the parasites using the Amaxa-Nucleofector™ as described above. Finally, allelic substitutions were confirmed by PCR amplification of the target gene using different combinations of primers. All the sequences are listed in the S1 Table.

### Evaluating the transcriptional regulation by RNA sequencing in the resistant clone

RNA from $2 \times 10^8$ epimastigotes was extracted using the *Quick-RNA* Miniprep Kit (Catalog number ZR1054, Zymo Research, California, USA) and sent to the University of Oklahoma (Oklahoma, USA) for library preparation using poly-A tails enrichment kit and sequencing on the Illumina NovaSeq 6000 platform with paired reads methodology. Ten paired-end samples corresponded to 61S and 61R_cl4, with three or two biological replicates, respectively, were used. The bioinformatic analyses were done as described elsewhere [23].

### Analysis of mutations in the NTRI gene between sensitive and resistant clones

The RNA-seq data obtained were aligned to the NTRI reference gene (C4B63_56g60) to select all reads belonging to this gene. Subsequently, the generated.sam files were converted, sorted, and indexed with the samtools software [24]. For variant calling, only reads with a sequencing quality greater than or equal to Q30 were used, and bcftools software was used to generate a depth and coverage profile from the aligned reads and identify SNPs and indels [25]. Finally, a consensus sequence was generated for each treatment, and its replicates were used to integrate the variants detected on the reference sequence.

### Comparative analysis with other *T. cruzi* resistant clones

To compare the expression profiles of different Bz-resistant *T. cruzi* clones, a comparative analysis was performed using RNA-seq data among our two populations (61S and 61R_cl4) and those previously reported by Lima et al. (2023) (WTS and LER) [26]. Data were analyzed as described above for our *T. cruzi* clones.

## Results

### Pressure with Bz generates *T. cruzi* clones with different resistance profiles

After pressing a sensitive clone (61S) for 20 passages, we got a phenotype that was 15.8 times more resistant than its parental clone (Fig 1A). We obtained 14 clones from this phenotype with multiple resistance profiles, from which we selected one clone (61R_cl4) with the highest resistance to Bz (Figs 1A and S2A). This clone also showed cross-resistance to Nfx (Fig 1B). The growth curve and the cell cycle results showed significant changes between the 61R_cl4 clone and 61S. Thus, it was observed that the first had a shorter doubling time of 23 hours, against 35 hours obtained in the latter (S2B Fig). These results were confirmed with the quantification of parasites in the different phases of the cell cycle, where it was observed that in the G1 phase, more than 57% of the cells of clone 61S were found, while only 43% of clone 61R_cl4. In addition, a more significant number of cells in the G2 phase (20% vs 12%) of the 61R_cl4 population compared to 61S (S2C Fig).

### Characterization of the NTRI gene in resistant parasites suggests mechanisms independent of this enzyme

Since the NTRI enzyme has been postulated as the most important mechanism of resistance *in vitro* and *in vivo* to Bz and Nfx in *T. cruzi* [10–12], we decided to characterize the expression and presence of mutations in the 61R_cl4 resistant clone. Analysis of NTRI expression by WB showed that clone 61R_cl4 had no under-expression of this protein, suggesting no loss of gene copies as resistance was gained (S3A Fig). On the other hand, to determine whether there were point mutations in the coding region, the sequences of the NTRI gene were analyzed for both clones 61S and 61R_cl4. After aligning the sequences against a reference sequence (C4B63_56g60), there were no mutations in this gene for either clone (S3B Fig). These results suggest that other resistance mechanisms independent of this enzyme are present in clone 61R_cl4.

### NTRI knockout induces resistance to Bz comparable to the 61R_cl4 clone

The results above suggest that NTRI is not involved in the resistance to Bz. To verify if this gene has some role in the resistance to Bz in our phenotype, we partially (SKO_NTRI) and fully knocked out (DKO_NTRI) the coding region of NTRI in the 61S_Cas9 parasites (S4 Fig). The results showed that NTRI indeed induces resistance to Bz and supports the idea that the resistant clone exhibits mechanisms other than NTRI. Interestingly, we observed that the resistance level generated in the NTRI knockout parasites (DKO_NTRI) was similar to that obtained in the resistant clone 61R_cl4 (Figs 1A and 1C).

### Gene expression analysis between Bz-sensitive and resistant clones reveals novel regulated genes

In order to search for new genes involved in the resistance to Bz, the transcriptome of the Bz-sensitive (61S) and resistant (61R_cl4) clones was analyzed. Initially, a principal component analysis was performed, which evidenced a clear separation between the control group (61S) and the experimental treatment (61R_cl4), suggesting different transcriptomic profiles and heterogeneity in the susceptible population (Fig 2A). The difference in expression profiles was confirmed with the heatmap constructed from the differentially regulated genes (DRGs) (Fig 2B). Of 890 DRGs (with false discovery rate (FDR)-adjusted *p* values less than 0.05 and a Log2 fold change greater than 0.58), 236 genes were positively and 654 were negatively regulated (Fig 2C; S1 File). Interestingly, none corresponded to NTRI, confirming our previous

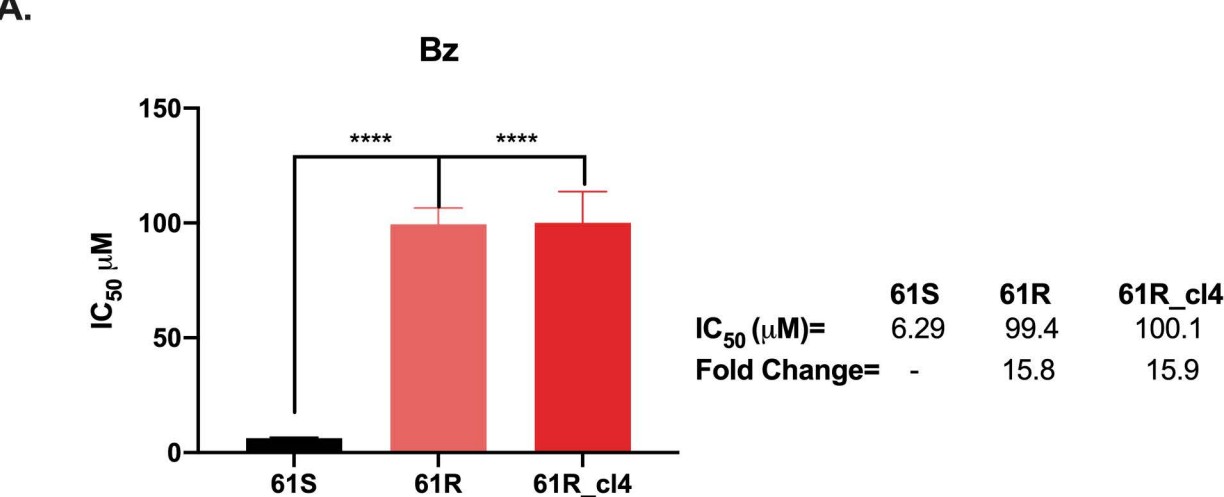

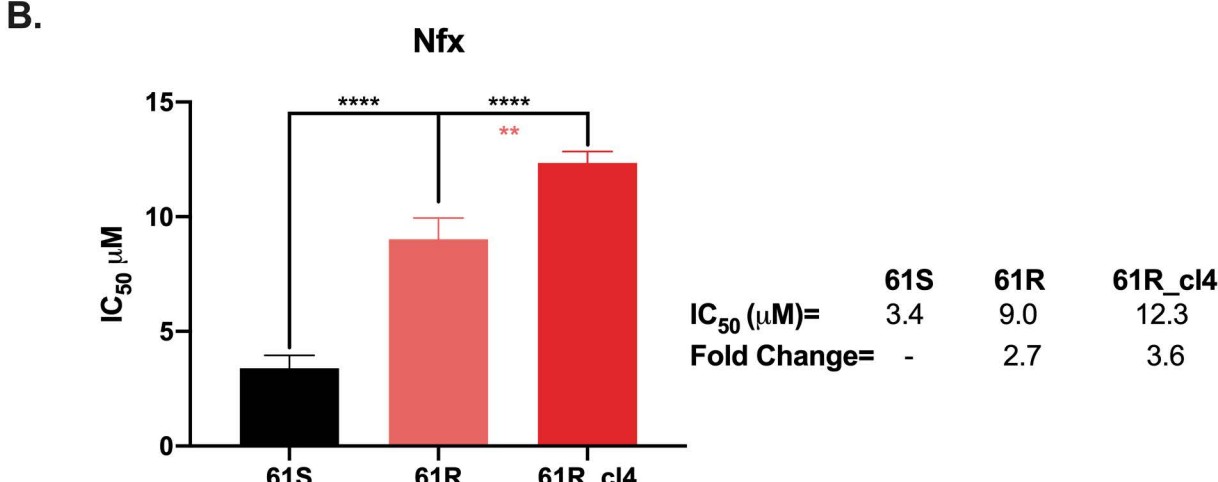

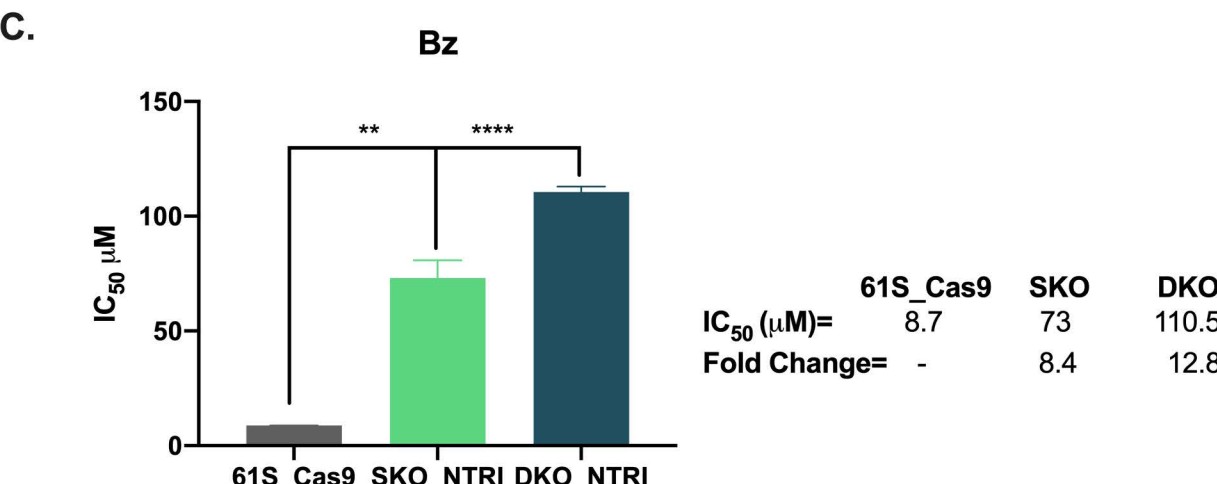

**Fig 1. Benznidazole resistance profiles of *Trypanosoma cruzi* clones.** $IC_{50}$ to Bz (**A and C**) and Nfx (**B**) were evaluated by alamarBlue for different clones in triplicate. 61S: sensitive clone; 61R: resistant phenotype; 61R_cl4: resistant clone isolated from 61R; 61S_Cas9: 61S clone

expressing Cas9 enzyme; SKO_NTRI: 61S_Cas9 clone with loss of one allele of the NTRI gene; DKO_NTRI: 61S_Cas9 clone with loss of both alleles of the NTRI gene. Statistical significance was determined in GraphPad Prism 8.0 using the one-way ANOVA with Dunnett's or Tukey's multiple comparison tests to Bz and Nfx, respectively. ****$p < 0.0001$; **$p < 0.01$. The fold change in the $IC_{50}$ was calculated with respect to control parasites (61S or 61S_Cas9).

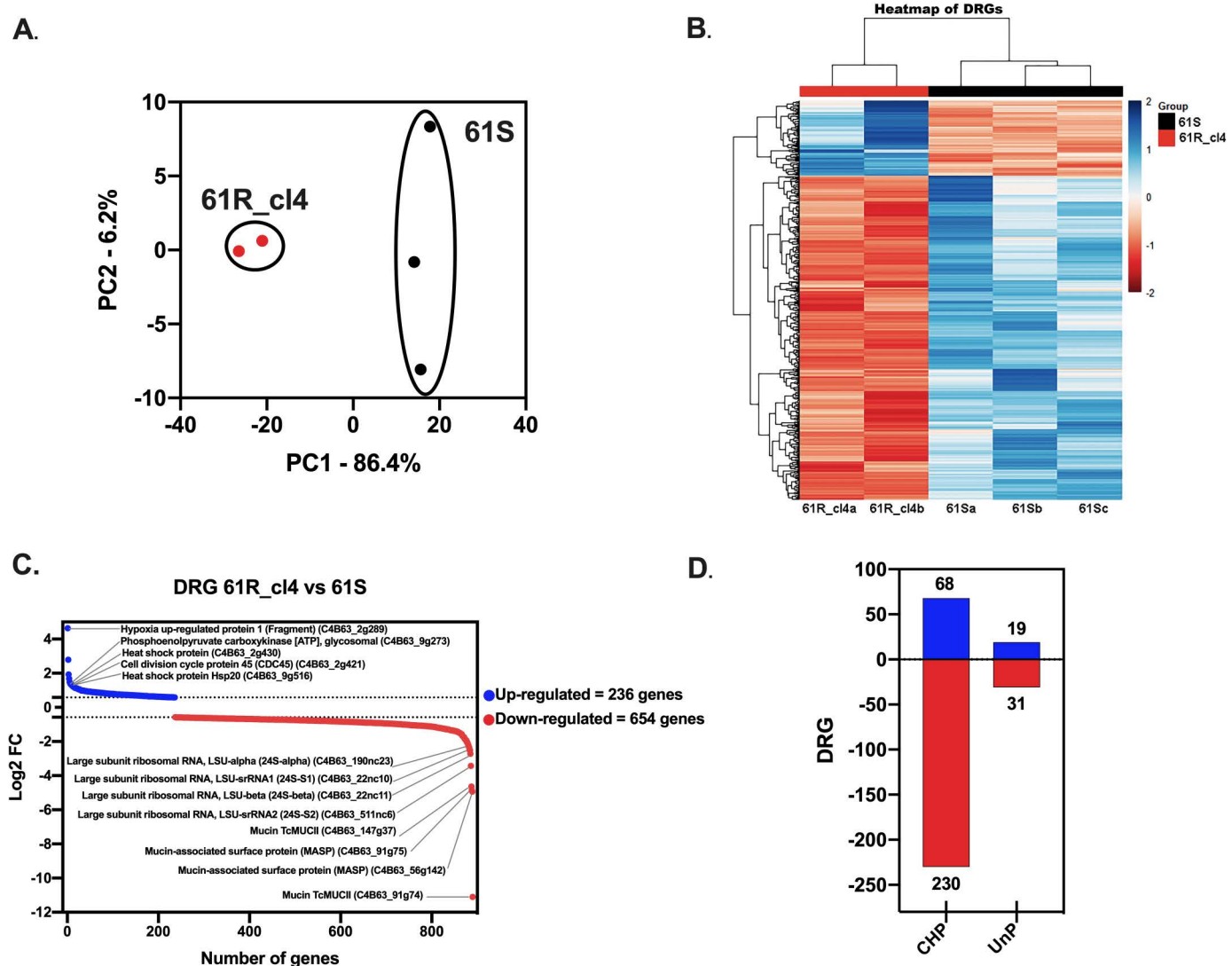

**Fig 2. Differentially Regulated Genes (DRGs) from Bz-sensitive and resistant clones of *Trypanosoma cruzi*. A.** PCA analysis of the RNA sequencing data from two or three biological replicates of 61R_cl4 and 61S, respectively. **B.** Analysis of differentially regulated genes (DRGs) from 61R_cl4 compared to 61S. **C.** Up-regulated (blue) and down-regulated (red) genes with the highest statistical significance. **D.** DRGs annotated as conserved hypothetical protein (CHP) and unspecified product (UnP) and their respective regulation: Up (blue) or down (red).

analyses. Among the positively regulated genes with the highest fold change were hypoxia up-regulated protein 1 (fragment) (C4B63_2g289) involved in the response to oxidative stress or low oxygen levels, heat shock protein (C4B63_2g430) and heat shock protein Hsp20 (C4B63_9g516) genes involved in response processes to stressful conditions, the cell division cycle protein 45 (CDC45) gene (C4B63_2g421) involved in DNA replication processes, and

the phosphoenolpyruvate carboxykinase [ATP] glycosomal gene (C4B63_9g273) involved in parasite energy metabolism. On the other hand, among the most fold chance negatively regulated genes were the ribosomal RNA components LSU-srRNA2 (C4B63_511nc6), LSU-beta (C4B63_22nc11), LSU-srRNA1 (C4B63_22nc10) and LSU-alpha (C4B63_190nc23) involved in the protein synthesis process, and surface genes belonging to the Mucin group (Fig 2C). 39.1% of the DRGs are annotated as conserved hypothetical protein (CHP) and unspecified product (UnP) (Fig 2D).

## Functional enrichment of DRGs reveals genes possibly associated with drug defense

To obtain additional information on the Bz response of the 61R_cl4 population, a gene ontology (GO) analysis was performed using TriTrypDB. For the negatively regulated DRGs, a total of 178 GO terms distributed in three domains were obtained: 85 belonging to biological processes, 72 to molecular functions, and 21 to cellular components (S2 File). Fig 3 shows the most specific GO terms, based on manual filtering. The top GO terms observed were related to transmembrane transport of a variety of substrates across cellular and subcellular membranes, finding 21 genes associated with ADP/ATP transport across the mitochondrial membrane, hexose transporters involved in cellular energy metabolism, nucleobase transport, pH regulation through sodium and hydrogen ion exchange, and genes involved in multidrug resistance that give the ability to expel toxic compounds out of the cell. Additionally, genes associated with sphingolipid biosynthesis and 2-oxoglutarate metabolisms, pathways within lipid metabolism, and the Krebs cycle, respectively, both essential for energy production and

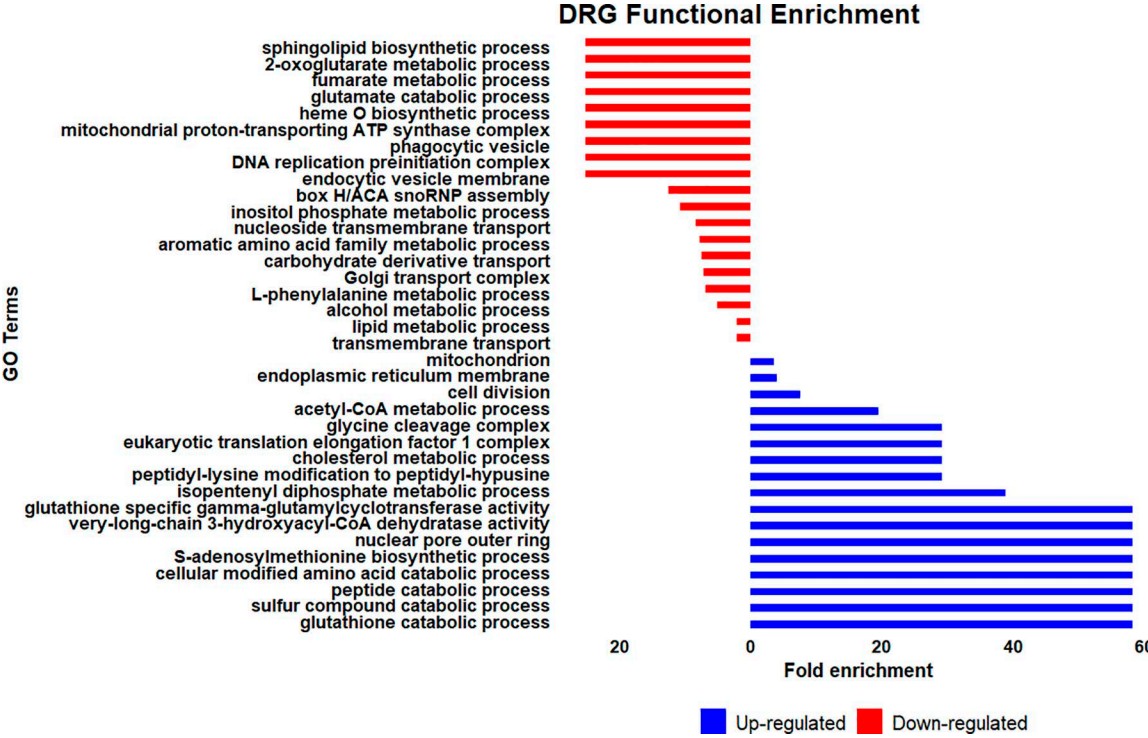

**Fig 3. Functional enrichment of Bz- resistant *Trypanosoma cruzi* clone Differentially Regulated Genes (DRGs).** The most specific GO terms, based on manual filtering are shown. Up-regulated (blue) and down-regulated (red). The size of the bar shows the fold enrichment for each GO term.

cellular signaling, were affected. At the level of cellular components, the analysis showed the affectation of the ribosome distinguishing genes associated with structural and functional elements, the mitochondrial ATP synthase and the golgi transport complexes, essential cellular components for protein assembly, and the generation of ATP (S2 File).

On the other hand, for positively regulated DRGs, 161 GO terms were obtained and distributed in three domains, 78 belonging to biological processes, 58 to molecular functions, and 25 to cellular components (S2 File). The GO analysis revealed that biological processes related to nucleobase metabolism, DNA metabolic processes, cell cycle, DNA replication, and DNA repair were affected. Moreover, genes acting in the catabolism of glutathione, sulfur compounds, and peptides involved in the degradation of molecules to recycle building blocks and maintain cellular redox balance were also regulated. Finally, cellular components involved in energy production and cellular metabolism were also affected.

### GO enrichment analysis shows the role of ABC transporters and proteins involved in DNA repair

A manual classification of different functional groups was performed to understand better the genes involved in the enrichment of GO terms. A total of 24 genes associated with transmembrane transport were found, of which 21 are negatively regulated. Interestingly, within this group, ABC transporter genes were identified as the case of multidrug resistance protein E (C4B63_328g10) (Fig 4A). Similarly, 20 genes belonging to subunits of the ribosomal complex were also diminished (Fig 4B). Conversely, the enrichment results confirmed that among the most abundant positively regulated genes were those associated with DNA replication processes, genetic recombination, and DNA damage repairs, such as cell division cycle protein 45 CDC45 (C4B63_2g421, C4B63_2g422), DNA recombination and repair protein RAD51 (C4B63_2g439), DNA topoisomerase III (C4B63_6g416), DNA repair protein (C4B63_142g10), meiotic recombination protein SPO11 (C4B63_31g223), DNA repair and recombination helicase protein PIF6 (C4B63_43g206), as well as genes involved in RNA transcription, processing, and modification such as fibrillarin (C4B63_207g2), proteinaceous RNase P2 (C4B63_2g382), tRNA modification enzyme (C4B63_278g5), tRNA (guanine-N(7)-)-methyltransferase (C4B63_2g347), lysyl-tRNA synthetase (C4B63_45g239), and tryptophanyl-tRNA synthetase (C4B63_10g517) (Fig 4C).

### Comparative transcriptomics reveals the presence of shared genes in benznidazole-resistant *T. cruzi* populations

A comparative analysis was performed using RNA-seq data from a Bz-resistant population obtained by Lima et al. (2023) [26] to depurate the many genes found in the DRG analysis. Initially, as described above for our Bz-resistant *T. cruzi* clone, an analysis of the NTRI gene sequence in the resistant LER clone was performed to verify the presence or absence of mutations. The results showed that the LER clone also has no mutations in the NTRI gene (S3B Fig), but it was down-regulated. Subsequently, principal component analysis was performed among the four populations (61S, 61R_cl4, WTS, and LER), which showed a clear separation between the two resistant populations (61R_cl4 and LER), suggesting different transcriptomic profiles between these two populations despite their Bz resistance status (Fig 5A). Differential expression analysis identified 1,530 DRGs for the LER population, of which 865 genes were positively and 665 were negatively regulated (Fig 5B; S1 File). A total of 149 genes were shared for the two populations. Remarkably, the functional enrichment evidenced standard cellular components such as mitochondria and ribosomes, important protein complexes such as the eukaryotic translation elongation factor 1 complex, and ribonucleoprotein complexes containing nucleolar small

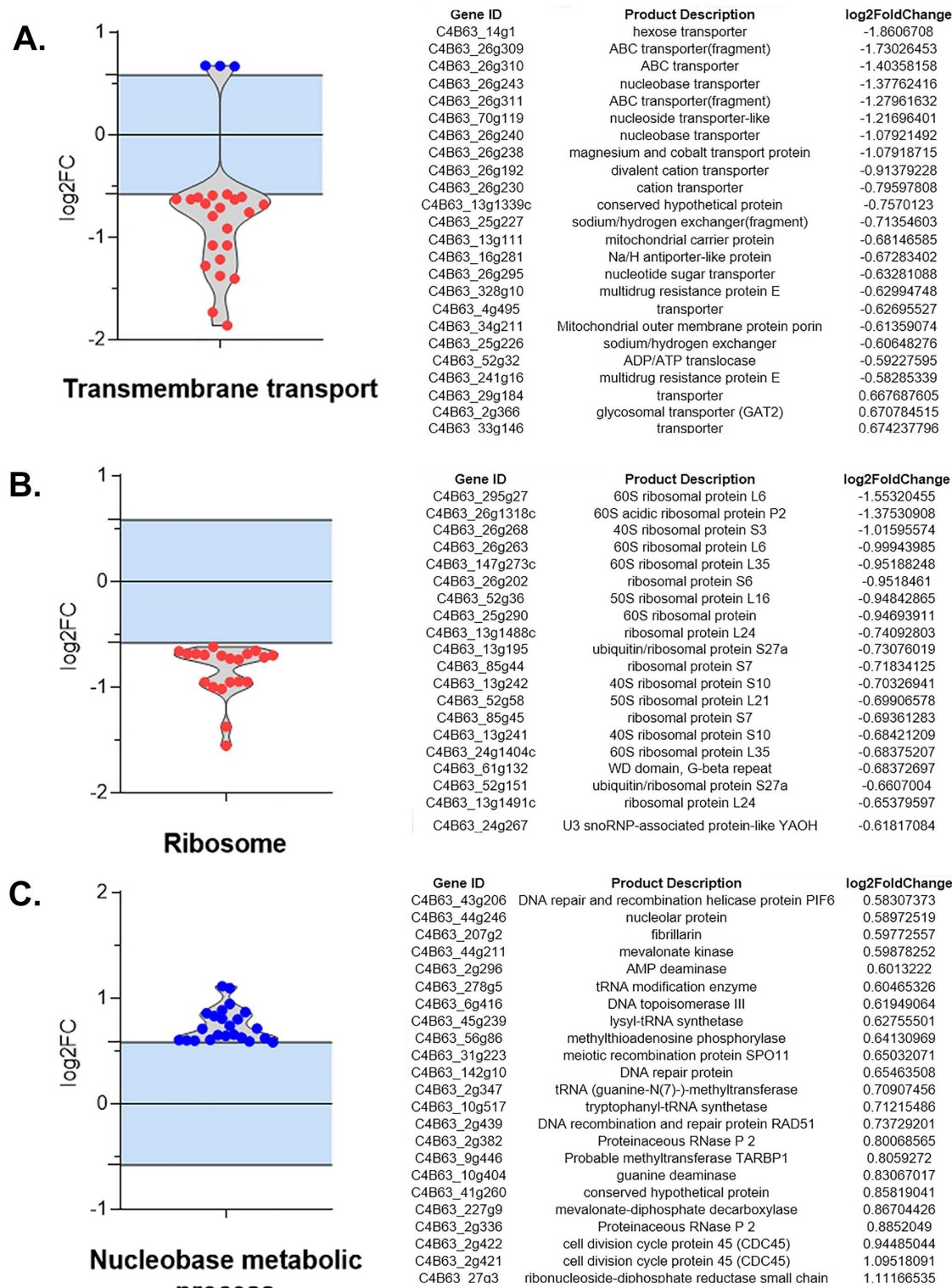

**Fig 4. Violin plots of DRGs genes within functional groups in Bz-resistant *Trypanosoma cruzi* clone.** Differential regulated genes associated with transmembrane transport (**A**), ribosome (**B**) and nucleobase metabolic process (**C**). On the left is shown the distribution of genes belonging to specific functional categories as a function of Log2 FC and their respective regulation: Up (blue) or down (red). On the right is a detailed description of the genes belonging to each functional category, their ID, and their Log2 FC value.

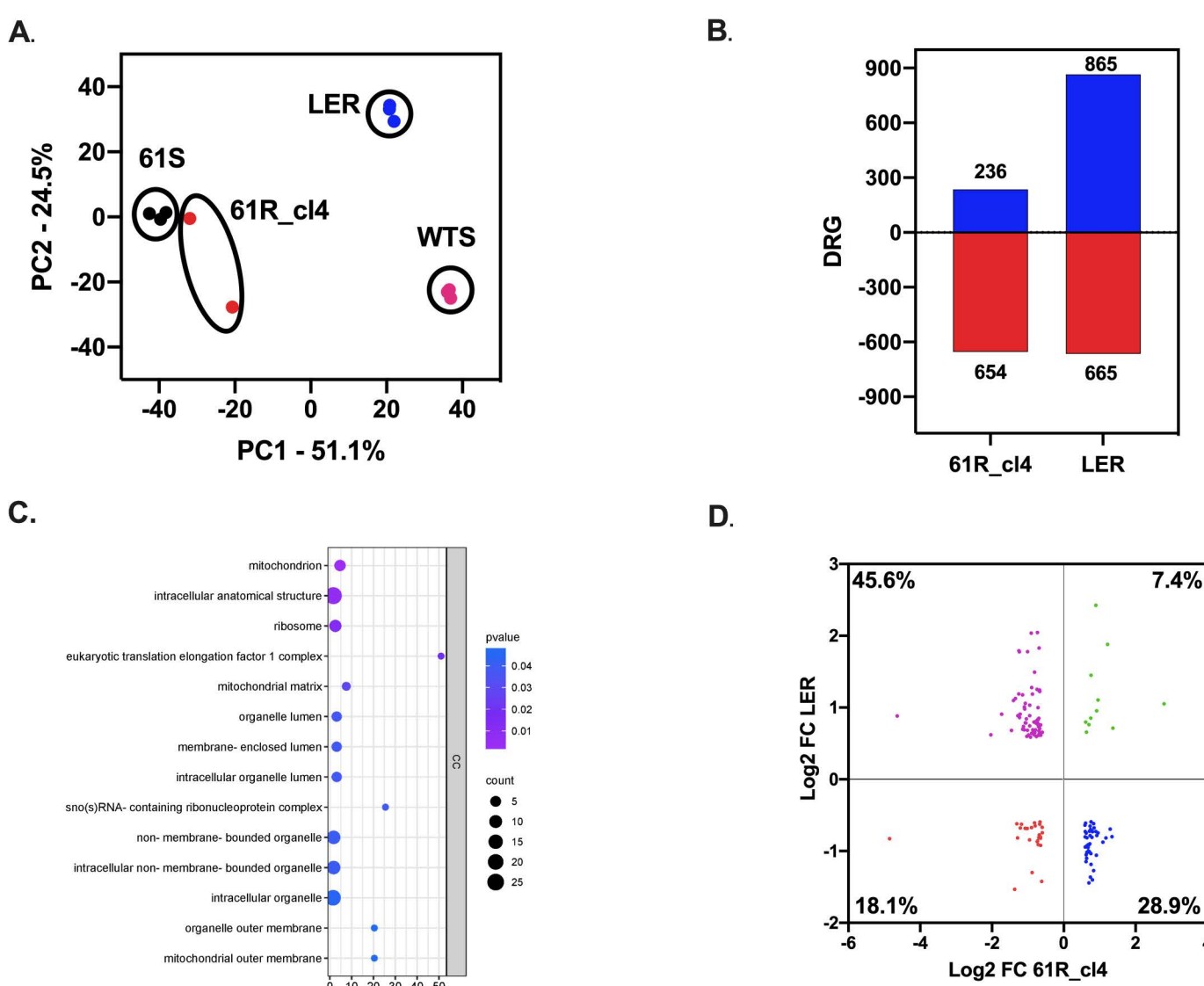

**Fig 5. Comparative transcriptomics between benznidazole-resistant *Trypanosoma cruzi* populations. A.** PCA analysis of the RNA sequencing data from 61R_cl4 and 61S, LER, and WTS *T. cruzi* clones. **B**. Comparison of differentially regulated genes (DRGs) among 61R_cl4 and LER *T. cruzi* Bz-resistant populations. **C**. Functional enrichment of genes shared DRGs between 61R_cl4 and LER populations. **D.** Scatter plot of the common DRG genes from 61R_cl4 and LER populations as a function of their Log2 FC.

RNAs (snoRNA) (Fig 5C). Of these common genes, 18.1% were found negatively regulated in the two populations, such as ribosomal subunits L24, S27a, S10, L35, S3, S7 (C4B63_13g1488c, C4B63_13g195, C4B63_13g241, C4B63_147g273c, C4B63_24g1404c, C4B63_26g268, C4B63_85g45), the fructose-bisphosphate aldolase glycosomal gene involved in glycolysis (C4B63_13g270, C4B63_13g271), the Hsp70-Hsp90 organizing protein 2 gene (C4B63_34g103), the proteolytic enzyme cysteine peptidase, clan CA, family C51 (C4B63_9g333), four mucin-rich surface proteins (C4B63_34g1428c, C4B63_34g216, C4B63_34g326, C4B63_91g75) and six conserved hypothetical protein (C4B63_220g10, C4B63_34g118, C4B63_34g1211c, C4B63_34g1250c, C4B63_34g292, C4B63_34g314). On the other hand, 7.4% of the genes in common positively regulated in the two populations, possess mucin-associated surface proteins (MASP) and trans-sialidases (C4B63_29g37, C4B63_74g13, C4B63_53g223, C4B63_74g138),

conserved hypothetical protein (C4B63_104g81, C4B63_234g2, C4B63_87g29), unspecified product (C4B63_1g197, C4B63_22g138, C4B63_64g141), and the CRAL/TRIO gene, N-terminal domain/CRAL/TRIO domain-containing protein (C4B63_18g39). Finally, 45.6% of the genes in common were positively regulated in the LER population and negatively regulated in the 61R_4 population, while 28.9% were positively in the 61R_cl4 population and negatively regulated in the LER population (Fig 4D, S3 File).

### DAVID analysis confirms significant down-regulation of ribosomal genes between resistant clones

As mentioned above, a considerable number of ribosomal subunits were found to be negatively regulated between the 61R_cl4 and LER populations. The results were confirmed when a DAVID (Database for Annotation, Visualization, and Integrated Discovery) enrichment analysis was performed (S5 Fig, S4 File). The results showed that proteins S3e (C4B63_26g268), L35e (C4B63_147g273c, C4B63_24g1404c), L24e (C4B63_13g1488c), S27Ae (C4B63_13g195) and S7e (C4B63_85g45) were shared in both populations. Meanwhile, the LP1-LP2 (C4B63_26g1318c), L21 (C4B63_52g58), L6 (C4B63_295g27, C4B63_26g263), and S10 (C4B63_13g241, C4B63_13g242) proteins were specifically found in the 61R_cl4 population and surprisingly 28 large subunit ribosomal proteins and 21 small subunit proteins were found only for the LER population (S5 Fig).

## Discussion

The emergence of parasite drug resistance represents a substantial threat to morbidity and mortality worldwide, complicating the treatment of infectious diseases and increasing healthcare costs. Successful treatment and disease control require the implementation of cutting-edge strategies that allow not only the identification of new bioactive molecules but also a holistic understanding of their mode of action, their target(s), and the strategies (genes and mechanisms) used by pathogens to resist drug action [27].

NTRI has been demonstrated to play a relevant role in resistance to Bz and Nfx *in vitro* in *T. cruzi* [10,11]. Furthermore, recent evidence suggests that it may contribute to therapeutic failure in patients treated with Bz, where different mutations may impact its activation [12]. Moreover, other enzymes have been identified as involved in Bz-resistant *T. cruzi* strains isolated from patients [15]. However, we must further understand which genes are essential in this process. It is, therefore, crucial to conduct transcriptomic, genomic, proteomic, and metabolomic studies comparing sensitive and resistant parasites. These approaches can facilitate the identification of shared regulated pathways and genes, offering potential targets for developing novel treatments.

This study used a comparative RNA-seq analysis to identify genes other than NTRI involved in benznidazole resistance in *T. cruzi*. In this regard, Mejia et al. (2012) reported that the emergence of Bz resistance in this parasite is associated with the loss of NTRI following increased drug pressure [10]. However, we have shown that prolonged exposure to a single high concentration of Bz over multiple generations leads to resistance via NTRI-independent mechanisms. It is also important to highlight that drug pressure was applied to a strain with different clones in the first study, some of which could be naturally resistant. In this work, on the other hand, we induced resistance in one clone, which could explain the differences between the resistance mechanisms in this parasite. These two approaches must be considered to understand the genes involved in drug resistance as they activate different biochemical pathways. These results also support that prolonged treatment regimens could potentially result in therapeutic failure due to the emergence of resistance.

It should be noted that the analysis of the *T. cruzi* transcriptome is still a challenge since, in addition to having many multigene families, it has many transcripts identified as conserved hypothetical protein (CHP) and unspecified product (UnP). In the first case, genes such as retrotransposon hot spot (RHS) protein, trans-sialidases, mucins, MASP, and dispersed gene family protein 1 (DGF-1) were found in the resistant clone with a decreasing trend in the number of transcripts. On the contrary, Lima et al. (2023) found more over-regulation of some of these genes [26]. These discrepancies may result from the type of genes and not from an actual difference between sensitive and resistant clones. However, it cannot be ruled out that chromosomal rearrangements are generated in the emergence of resistance, which may lead to the loss or gain of multicopy genes. The difference in the number of copies in these families' genes could affect the parasites' fitness, which has been widely demonstrated in different organisms [28–30].

Among the positively regulated genes with the most remarkable change in expression are some heat shock proteins (HSPs), which have been described to be involved in several critical cellular functions, especially during stress response, by regulating the stability and folding of proteins [31,32]. In *Leishmania*, these proteins are crucial for survival, confer thermotolerance, enhance virulence, and improve the parasite's ability to adapt to the hostile environment by allowing it to detect abrupt changes in its environment [33]. In *Trypanosoma brucei*, it has been proposed that these proteins ensure the correct folding, maturation, and degradation of essential proteins for the parasite, especially when faced environmental stresses, such as drug treatment [34]. Similarly, these proteins could be crucial in *T. cruzi* to resist oxidative stress induced by benznidazole pressure, explaining the over-regulation of several of them.

Interestingly, several down-regulated zinc finger proteins were observed (5 in total; S1 File). Recently, Queffeulou et al. (2024) demonstrated through functional studies that a loss-of-function of this gene contributes modestly to reduced susceptibility to MF in *L. infantum*. This may be because the gene plays a crucial role in ubiquitination, thereby influencing the stability and function of various proteins either through protein degradation or differential localization [22]. The role of these proteins in Bz resistance must be studied deeply in *T. cruzi*.

On the other hand, we highlight the CDC45 gene, which has been described as an essential protein for DNA replication. This gene interacts with critical components of the replication complex, such as proliferating cell nuclear antigen (PCNA), which acts as a scaffold for DNA replication factors. The interaction between CDC45 and PCNA through the PIP (proliferating cell nuclear antigen interacting protein) box is critical, as it enables the correct formation of the replication complex, thus ensuring efficient DNA replication [35]. In *T. cruzi,* in the context of Bz resistance, cells may rely on enhanced DNA replication and repair mechanisms to counteract drug-induced oxidative stress [36]. Positive regulation of CDC45 in resistant clones could favor more efficient DNA replication and repair under Bz genotoxic stress, helping the parasite maintain genome integrity and survive the treatment [26]. In this sense, this result was partially corroborated with the cell cycle and growth curves experiments, where a faster replication of the resistant clone was observed compared to the sensitive one. This can also be explained by several up-regulated genes involved in the cell cycle and DNA repair processes. On the other hand, high DNA replication has been associated with the accumulation of mutations in different microorganisms, some of which may favor the emergence of drug resistance [37,38].

Other highlighted enzymes are phosphoenolpyruvate carboxykinase (PEPCK) and trypanothione synthetase (TRYS), found overregulated in the resistant clone. The first plays a crucial role in metabolic processes such as glycolysis and gluconeogenesis. In *T. cruzi*, the glycosomal form of PEPCK is essential for maintaining energy balance, particularly under stress conditions. This metabolic flexibility could allow parasites to survive better in the presence of drugs.

PEPCK ensures a constant supply of metabolic intermediates necessary for cellular repair and stress resistance [39]. TRYS catalyzes the biosynthesis of trypanothione, the major thiol of trypanosomatids, which, in its reduced form, is involved in various functions such as detoxification [40]. In our resistant parasites, more biosynthesis of this thiol may help to accelerate detoxification processes.

Transcriptomic analysis showed a considerable number of negatively regulated ribosomal subunits, which may suggest an essential pathway in drug resistance based on ribosome depletion that could generate a reduction in protein synthesis, minimizing the metabolic burden during drug exposure and redistributing resources towards other processes essential for resistance. It has been evidenced that alterations in ribosomal subunit levels play a crucial role in drug resistance [41]. During the last few years, the concept of specialized ribosomes has been put forward, which presents the idea that ribosomes actively participate in the regulation of translation and have a specialized response to different types of stimuli [42].

Additionally, it was recently demonstrated that multiple ribosomal proteins containing cysteine residues are strongly regulated by anticancer agents that have been shown to increase steady-state ROS levels. These cysteine residues were postulated as potential ribosomal sensors of ROS imbalance [43]. In our work, we obtained several ribosomal proteins significantly regulated when the parasites were exposed to benznidazole, a prodrug that also produces alterations in ROS. Surprisingly, some cysteines were found when we looked for cysteine residues in these ribosomal proteins. For instance, the S3 ribosomal protein contains the cysteine 71 (C71); S7 (C138, C147); S27 (C33, C43, C45, C120, C125, C140), and L24 (C6, C40). Interestingly, these proteins were regulated in the resistant parasites obtained in our study and those reported previously by Lima et al. (2023) [26]. The role of these cysteines in the response to benznidazole and ROS in *T. cruzi* should be studied in future investigations.

One of the essential contributions of our study was the identification of common regulated genes by comparative transcriptomics with those obtained by Lima et al. (2023) [26]. Although the levels of resistance between both clones are different, and the NTRI enzyme seems to affect the resistance of the LER clone as it is downregulated, some ribosomal genes appear among the commonly regulated genes. This could reinforce the idea of the importance of these genes in resistance and the need for future studies aimed at characterizing them in terms of resistance.

Our findings highlight the complexity of Bz and Nfx resistance in *T. cruzi*, suggesting that resistance may manifest as either a monogenic or polygenic trait. In some cases, resistance may be monogenic, primarily through downregulation or point mutations in the NTRI gene. This impairs the activation of these nitro drugs and confers a resistance phenotype. This single-gene mechanism is supported by previous studies showing that loss-of-function mutations or deletions in NTRI can lead to resistance [10,11]. However, a recent study using Mexican *T. cruzi* isolates with different Bz and Nfx resistance profiles showed by digital PCR that resistant parasites had increased NTRI expression. This was counterbalanced by overexpression of detoxifying enzymes such as SOD and GTS or transporters such as MDR [44]. Our transcriptomic data also suggest a polygenic model, where the cumulative effect of multiple pathways drives resistance. This model posits that genes involved in oxidative stress response, drug transport, protein synthesis, and DNA repair are co-regulated, contributing to an enhanced resistance phenotype in a multi-factorial manner. Such a polygenic trait could reflect a broader adaptive response by *T. cruzi*, allowing the parasite to mitigate drug-induced damage through a network of complementary resistance mechanisms. Recognizing both monogenic and polygenic resistance pathways is crucial for developing targeted strategies against drug-resistant strains. It suggests therapeutic approaches must address multiple targets to combat resistance effectively.

Overall, our results support a multigenic response to benznidazole in *T. cruzi*. We propose a mechanism where NTRI is crucial in the mechanism of drug resistance in the parasite by the activation of nitro drugs such as benznidazole and nifurtimox. However, other genes are regulated in the parasite when this enzyme is active. Thus, once the drugs are activated, the metabolite glyoxal is produced, which induces damage in macromolecules. In response to glyoxal, the parasite overexpresses other genes like AKR and ADH, which have been reported as glyoxal detoxifying enzymes [15]. Additionally, *T. cruzi* could respond by overexpressing different genes, mainly involved in oxidative stress, energetic metabolism, cell division, DNA repair, and ABC transporters. The toxic stress produced also affects the cell cycle because the parasite is forced to respond to the drug, subsequently diminishing ribosomal protein production. Remarkably, genes involved in these processes were found to be regulated in our work. A summary of our most relevant findings regarding to differentially regulated genes in a Bz-resistant *T. cruzi* clone is depicted in Fig 6.

## Conclusions

Our results support the idea that resistance to benznidazole is a multigenic trait. The parasite activates genes involved in drug activation, elimination of secondary metabolites, DNA repair, and replication mechanisms. Genes involved in response to oxidative stress, energy metabolism, and membrane transporters also help counteract the drug's effects. Deep knowledge of these genes is essential for designing new drugs to treat Chagas disease.

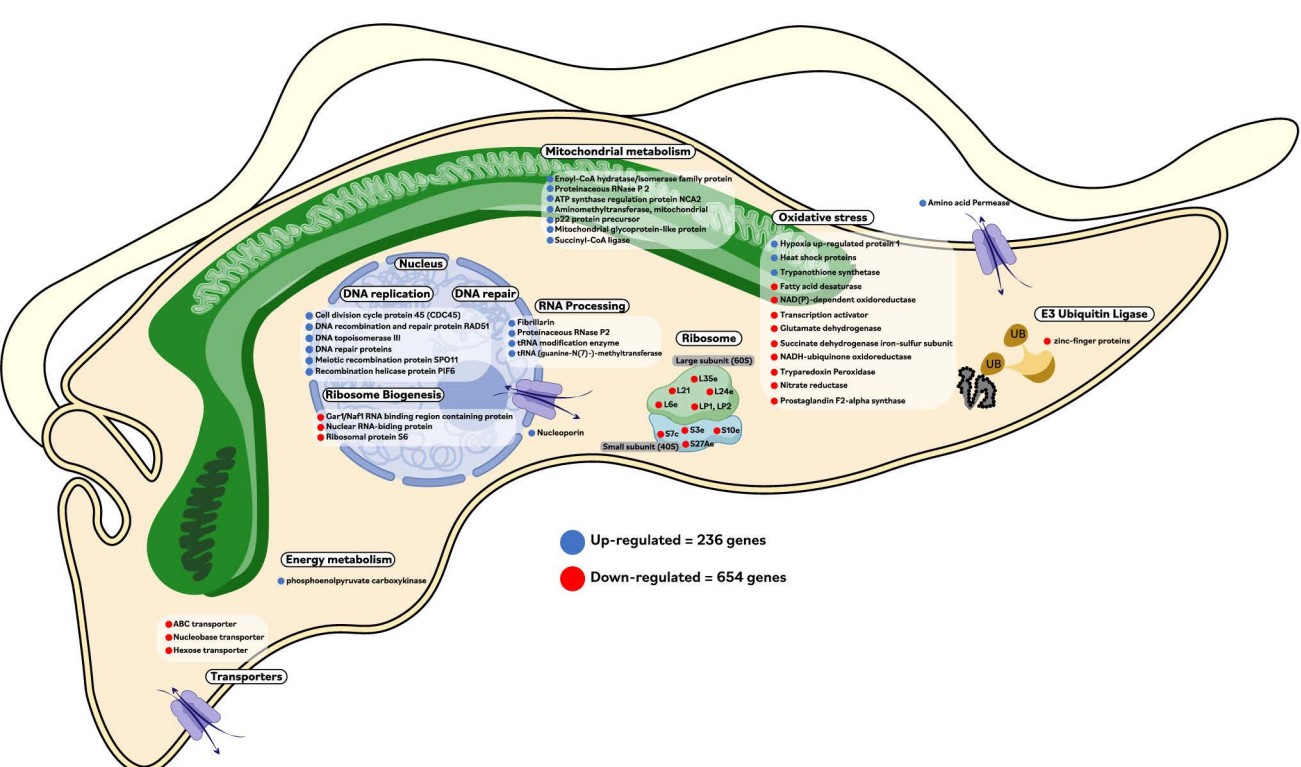

**Fig 6. Summary of the most relevant biochemical pathways and Differentially Regulated Genes (DRGs) in a Bz-resistant *Trypanosoma cruzi* clone.** Up-regulated (blue) and down-regulated (red).

## Supporting information

**S1 Fig. General workflow of the methodology implemented in this work. 1.** To generate benznidazole resistance from the 61S susceptible clone epimastigotes were subcultured every week under selective pressure until a resistant clone (61R_cl4) was isolated. **2.** RNA from both clones was extracted and sequenced using Illumina NovaSeq 6000 platform with paired reads methodology. **3.** The quality of the reads obtained was evaluated and the primers, adapters and sequences with a Phred value lower than Q30 were removed. **4.** Reads were individually aligned against the *T. cruzi* Dm28c reference genome (2018) from DTU TcI. **5.** Differential expression analysis was performed.
(TIF)

**S2 Fig. Biological characterization of the resistant clone. A.** The growth inhibition percentage of *T. cruzi* parasites at Bz was evaluated by alamarBlue and calculated by GraphPad Prism 8.0 from one sensitive clone (61S; black), one resistant phenotype (61R; pink), and 14 resistant clones (gray). In red is highlighted the clone with the lowest inhibition percentage to the drug (61R_cl4; the most resistant). The fold change in the $IC_{50}$ was calculated with respect to control parasites (61S). **B.** Epimastigotes proliferation curves were assessed by counting in the Neubauer chamber every 24 hours for ten days. Statistical significance was determined using two-way ANOVA with Sidak´s multiple comparisons test, performed in GraphPad Prism 8.0. ****$p < 0.0001$; ***$p < 0.001$; **$p < 0.01$. Sensitive parasites (61S) are represented in black, and resistant parasites in red (clone: 61R_4). **C.** Flow cytometry analysis of cell cycle progression. The 61S and 61R_cl4 populations were synchronized with HU for 16 h, and readings were performed using a BD LSRFortessa Cell Analyzer flow cytometer. The graphs showed the median fluorescence intensity of propidium iodide (PI).
(TIF)

**S3 Fig. Analysis of expression and point mutations in the NTRI gene in the susceptible and resistant parasites. A.** Expression level of NTRI gene from sensitive and resistant parasites determined by WB using 100 µg of total proteins from each clone. The intensity of the bands was quantified in the Odyssey Classic Infrared System. **B.** Aminoacid alignment from NTRI protein deduced from DNA sequencing of different *T. cruzi* clones. C4B63_56g60: reference sequence; 61S: susceptible clone; 61R_cl4: resistant clone; LER and WTS correspond to *T. cruzi* resistant and susceptible clones, respectively, obtained by Lima et al., 2023 [26].
(TIF)

**S4 Fig. DNA construct and confirmation by PCR of NTRI single and double knockout in *Trypanosoma cruzi*.** Schematic representations of different PCRs were performed to confirm the knockout of NTRI and the integration of the puromycin (PURO) gene in the NTRI locus. The agarose gel shows the result of the PCR of control parasites (Cas9) (1) and single (2) or double knockout (KO) (3) obtained after the transfection with gRNAs for NTRI gen (above) and puromycin HRTs (below) in the pTREX/Cas9 parasites. All the primer sequences (a, b, c, and d) are listed in the S1 Table .
(TIF)

**S5 Fig. Ribosomal profile (KO03010).** Negatively regulated ribosomal subunits in the 61R_cl4 (red stars) and LER (blue stars) in *Trypanosoma cruzi* Bz-resistant populations.
(TIF)

**S1 Table. Sequence of gRNA and primers used in the knockout of *Trypanosoma cruzi* NTRI gene.**
(XLSX)

**S1 File. Read counts and Differentially regulated genes (DRGs).** Read counts obtained from Bz-sensitive and resistant *Trypanosoma cruzi* clones. 61R_cl4: resistant clone; 61S: sensitive clone; LER and WTS: resistant and sensitive clones, obtained by Lima et al. (2023) [26]. The letters a, b, and c correspond to the replicates (Sheet ReadCounts). DGR between Bz-sensitive (61S) vs. Bz-resistant (61R_cl4) *Trypanosoma cruzi* clones (Sheet 61Svs61R) and DGR between Bz-sensitive (WTS) vs. Bz-resistant (LER) *Trypanosoma cruzi* clones (Sheet WTSvsLER). Up- and down-regulated genes are highlighted in blue and red, respectively. (XLSX)

**S2 File. Gene Ontology annotation of up- and down-regulated genes from 61R_cl4 clone.** BP: Biological process; CC: Cellular component; MF: Molecular Function. (XLSX)

**S3 File. Gene Ontology annotation of up- and down-regulated genes from LER clone.** BP: Biological process; CC: Cellular component; MF: Molecular Function. (XLSX)

**S4 File. DAVID enrichment analysis between 61R_cl4 and LER-resistant *Trypanosoma cruzi* clones.** (XLSX)

**S5 File. Minimal data set.** (ZIP)

## Author contributions

**Conceptualization:** Ana Maria Mejia-Jaramillo, Omar Triana-Chávez.

**Data curation:** Hader Ospina-Zapata, Geysson Javier Fernandez.

**Formal analysis:** Ana Maria Mejia-Jaramillo, Hader Ospina-Zapata, Geysson Javier Fernandez.

**Funding acquisition:** Ana Maria Mejia-Jaramillo, Omar Triana-Chávez.

**Investigation:** Ana Maria Mejia-Jaramillo, Hader Ospina-Zapata, Omar Triana-Chávez.

**Methodology:** Ana Maria Mejia-Jaramillo.

**Project administration:** Omar Triana-Chávez.

**Supervision:** Ana Maria Mejia-Jaramillo, Omar Triana-Chávez.

**Writing – original draft:** Ana Maria Mejia-Jaramillo.

**Writing – review & editing:** Ana Maria Mejia-Jaramillo, Hader Ospina-Zapata, Geysson Javier Fernandez, Omar Triana-Chávez.

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
