## [Decision Letter · Decision Letter 0]

17 Dec 2024

PONE-D-24-50847Transcriptomic analysis of benznidazole-resistant Trypanosoma cruzi clone reveals nitroreductase I-independent resistance mechanismsPLOS ONE

Dear Dr. Ana Maria Mejia-Jaramillo,

Thank you for submitting your manuscript to PLOS ONE. After careful consideration, we feel that it has merit but does not fully meet PLOS ONE’s publication criteria as it currently stands. Therefore, we invite you to submit a revised version of the manuscript that addresses the points raised during the review process.

We look forward to receiving your revised manuscript.

Kind regards,

Claudia Patricia Herrera, Ph.D.

Academic Editor

PLOS ONE

Journal Requirements:

“This work was supported by Universidad de Antioquia UdeA, SGR Grant BPIN 2020000100479.”

Additional Editor Comments (if provided):

From the perspective of drug resistance, is there a possible correlation between different DTUs or strains of the parasite and their levels of resistance or susceptibility?

What are the genotypes of the strains used in this study? Discussing these aspects could help clarify this issue.

Have the researchers observed any variations in parasite susceptibility based on its biological stage? While epimastigotes are commonly utilized for studies, could trypomastigotes (the infective form in mammals) exhibit different responses? Additionally, have the researchers detected any differences at the transcriptomic level?

Line 88. Please clarify what you mean by "same experimental and environmental conditions."

Line 198. It would enhance the discussion by providing more information about the strains and clones analyzed.

Line 462. Please replace "Trypanosoma cruzi" with "T. cruzi."

Figures: Please enhance the quality of the figures.

Reviewers' comments:

Reviewer's Responses to Questions

**Comments to the Author**

1. Is the manuscript technically sound, and do the data support the conclusions?

Reviewer #1: Yes

Reviewer #2: Yes

Reviewer #3: Yes

2. Has the statistical analysis been performed appropriately and rigorously? 

Reviewer #1: Yes

Reviewer #2: Yes

Reviewer #3: Yes

3. Have the authors made all data underlying the findings in their manuscript fully available?

Reviewer #1: Yes

Reviewer #2: Yes

Reviewer #3: Yes

4. Is the manuscript presented in an intelligible fashion and written in standard English?

Reviewer #1: Yes

Reviewer #2: Yes

Reviewer #3: Yes

5. Review Comments to the Author

Reviewer #1: The article presents a very novel work. It is approached from a transcriptomic approach to elucidate the possible mechanisms of resistance to BZ in populations of T. cruzi resistant by pressure to induced BZ combined with CRISPR Cas technology for genetic knock out (It was lacking to exploit these results a little more in the discussion). However, the discussion of the results needs to be strengthened in general by comparing with more related literature. Although one of the most relevant conclusions of the article is to reaffirm what has been reported by other authors, NTR I independent and multifactorial resistance mechanisms. It remains to further nurture this discussion by trying to present the possible alternative mechanisms that were rightly mentioned in the introduction (lines 44-50).

>What is the reason why the authors performed the work with BZ and not with NFX or both (not bad, but to know their reasons would be good). Why were IC50 determinations made for both, but only selective pressure against BZ? What is the reason for making this decision?

In the methodology it is mentioned that 10 concentrations of Bz and NFX were used, specify which were those concentrations. It also does not specify the presentation in which NFX and BZ were used, pure, tablet or purified from a tablet?

>What is the justification for performing the determination of the IC50 at 72 h?

>Specify in the methodology what was the concentration of rezarzurin used since only 20 ul/well is mentioned, but this is very ambiguous.

>How do they justify the use of the Epimastigote form and not the trypomastigote form?

>While the focus of the article is not to relate this to intra and inter DTUs genetic variability of T. cruzi (suggested to mention this in the introduction as a possible factor that may influence heterogeneity of response to Bz), how scalable or representable are these results for the other DTU's? Would one expect a big change using another DTU or another TcI isolate?

Reviewer #2: This is the revision of the study entitled 'Transcriptomic analysis of benznidazole-resistant Trypanosoma cruzi clone reveals nitroreductase I-independent resistance mechanisms' by Mejia-Jaramillo et al. The study is robust and well-executed, with the generation of benznidazole-resistant T. cruzi aligning with findings reported in the literature. While the study has some flaws, its findings are valuable to the scientific community as they highlight resistance mechanisms entirely independent of nitroreductase I.

The introduction includes a few ambiguous statements that require clarification. Additionally, the Materials and Methods section lacks crucial details necessary for reproducibility. The Results section also needs refinement to improve the clarity and presentation of the data.

Introduction

Lines 38-43: The end of this paragraph is highly inaccurate. Both Benznidazole and Nifurtimox have satisfactory results in the acute phase of Chagas disease (50-80% efficacy) but debatable efficiency in the chronic phase. The cited percentage of people discontinuing treatment is also unknown to me, as the last published finalized clinical trial BENEFIT (10.1056/NEJMoa1507574) showed a number of 13.4% of treatment discontinuation due to adverse events. If a study showed 20% of treatment discontinuation, please cite it in the text. I would cite clinical trials aiming to test both drugs in a prolonged treatment scheme too (10.1136/bmjopen-2021-052897).

I suggest updating the references and citing more relevant bibliography.

Here are some suggestions:

10.1111/bcp.14700

10.1056/NEJMct1014204

10.3390/vaccines12080870

Lines 45-47: “In resistance to Bz and Nfx, the enzyme nitroreductase I (NTRI) has been implicated as the primary gene responsible for resistance in vitro [4,5] and in patients with treatment failure”. The sentence is confusing; I would rephrase it to make it understandable that it is the gene encoding the enzyme nitroreductase that is associated with drug resistance.

Materials and Methods

Please revise the text to provide catalog numbers, the manufacturer of the kits/reagents used, and complete names and models of the equipment used.

The methods section is quite detailed and dense, which may make it challenging for readers to follow. To enhance clarity and accessibility, I recommend including a visual graphic that summarizes the key steps and analytical processes described in the paper. This could help readers better understand the workflow and overall approach at a glance.

Lines 74-81: In this section, it is not clear to me if you added Bz, G418, and Puromycin in the same media or if they were added into different media comprising different groups (one was treated only with Bz, one only with G418 and one only with Puromycin). Please clarify and explain why the use G418 and Puromycin (positive/negative controls? Or selection of a specific genotype?) If they’re for different experiments, please include the methodology for each experiment separately.

Are M-RATTUS, CO, 91, GAL-80 and 61.SUC different T. cruzi genotypes or clones? Please specify in the text, it is not clear. In that case, was the IC50 for Bz exactly the same value for all strains?

Lines 90-91: Cite a reference for the alamarBlue method for the determination of IC50.

Lines 101-102: From my understanding, only the 61.SUC clone was selected for Benznidazole pressure experiments, but that is not clear in the text. Please clarify how that decision was made and what led you to that decision.

Line 152: The numbers do not need to be written; they can be used as numerical numbers.

Lines 161: Include the name of the kit used for library preparation.

Results

Overall, if it is possible, I would change the colors to red for up-regulated and blue for down-regulated genes. This is what we most see in omics papers, and I believe it would make it easier for understanding.

Line 187: It’s ‘passages’ instead of ‘passes’.

Lines 187-198: Please mention the IC50 for each of the groups, corroborating fold-change resistance to Bz. If you mentioned it for Bz, mention the values for Nfx as well.

The 61S_Cas9, SKO_NTR1, and DKO_NTRI groups were not mentioned in the results, only in the figure legend. You either exclude this data from the graph or present the results in your text.

Lines 223-229: I see that the Figure 1 61S_Cas9, SKO_NTR1, and DKO_NTRI groups are mentioned here. However, I suggest separating the graphs and making another figure (or maybe Figure 1C) to fit this part of the results.

Line 240: Why was the Log2FC chosen of 0.58, when usually the Log2FC chosen for genes is 1?

Line 252: What is the importance of showing that the DRGs are annotated as conserved

hypothetical protein (CHP) and unspecified product (UnP)?

Figure 2C: Please increase the font size for the gene names, as they will be unreadable in the paper.

Figure 4. I recommend separating each functional group in (A) Transmembrane Transport, (B) Ribosome and (C) Nucleobase metabolic process, and mentioning it separately in the text.

S4 Fig. Legend description and presentation of results need to be improved. What does the blue, pink and yellow colors in the subunits mean? What question you plan to answer with this analysis?

Figure 6. Please increase the font size for readers.

Reviewer #3: The manuscript by Mejia-Jaramillo et al.describes the results of an exhaustive study centered in the characterization by RNA-seq analysis to identify genes other than nitroreductase-I (NTRI) involved in benznidazole and nifurtimox activity against Trypanosoma cruzi, the causative agent of American Trypanosomiasis (Chagas disease). The authors conclude that besides intrinsic NTRI toxicity, upregulation of heat shock proteins (HSPs) and of CDC45 in resistant clones could favor more efficient detoxification, DNA replication and repair under Bz/Nfx genotoxic stress. These are novel and very relevant findings in the characterization of molecular mechanisms involved in the resistance to the only two drugs currently available for the etiological treatment of Chagas disease, in terms of the potential prevention of such resistance via combination treatments.

6. PLOS authors have the option to publish the peer review history of their article (what does this mean? ). If published, this will include your full peer review and any attached files.

**Do you want your identity to be public for this peer review?** For information about this choice, including consent withdrawal, please see our Privacy Policy .

Reviewer #1: No

Reviewer #2: **Yes: ** Priscila Silva Grijo Farani

Reviewer #3: **Yes: ** Julio A. Urbina

---

## [Author Response · Author response to Decision Letter 1]

30 Dec 2024

December 29th 2024

Claudia Patricia Herrera, Ph.D.

Academic Editor

PLOS ONE

Dear Editor:

We thank the editor and reviewers for taking the time to review our manuscript PONE-D-24-50847. Their comments have undoubtedly improved the quality of our manuscript. We are glad that they found our manuscript of interest, and our responses to their comments are provided below. In the new version of the manuscript, the changes suggested by the editors and reviewers are indicated in red. Please also note that the order of the Supplementary Figures has changed in this version, as new data have now been included.

Sincerely,

Prof. Ana María Mejía Jaramillo

Universidad de Antioquia

Medellín, Colombia

General changes

Journal Requirements:

R/ The manuscript meets PLOS ONE´s style requirements.

“This work was supported by Universidad de Antioquia UdeA, SGR Grant BPIN 2020000100479.”

R/ The information was included in the Funding Statement and cover letter.

R/ I will make the data available.

4. PLOS requires an ORCID iD for the corresponding author in Editorial Manager on papers submitted after December 6th, 2016.

R /The ORCID iD was included.

Additional Editor Comments:

From the perspective of drug resistance, is there a possible correlation between different DTUs or strains of the parasite and their levels of resistance or susceptibility?

R/ Despite many efforts made by our research team and other researchers, we have not found a clear correlation between drug resistance and the DTUs. We have analyzed the IC50 to Bz of many strains from different DTUs, and there is no clear pattern between these two features. The following references showed the mentioned above.

● Gómez-Palacio, A., Lopera, J., Rojas, W., Bedoya, G., Cantillo-Barraza, O., Marín-Suarez, J., Triana-Chávez, O. & Mejía-Jaramillo, A. (2016). Multilocus analysis indicates that Trypanosoma cruzi I genetic substructure associated with sylvatic and domestic cycles is not an attribute conserved throughout Colombia. Infection, Genetics and Evolution, 38, 35–43. https://doi.org/10.1016/j.meegid.2015.11.026

● Mejia, A. M., Hall, B. S., Taylor, M. C., Gómez-Palacio, A., Wilkinson, S. R., Triana-Chávez, O., & Kelly, J. M. (2012). Benznidazole-resistance in Trypanosoma cruzi is a readily acquired trait that can arise independently in a single population. The Journal of Infectious Diseases, 206(2), 220–228. https://doi.org/10.1093/infdis/jis331

● Mejia-Jaramillo, A. M., Fernández, G. J., Montilla, M., Nicholls, R. S., Triana-Chávez, O. (2012). Sensibilidad al benzonidazol de cepas Trypanosoma cruzi sugiere la circulación de cepas naturalmente resistentes en Colombia. Biomédica, 32, 196-205.

What are the genotypes of the strains used in this study? Discussing these aspects could help clarify this issue.

R/ The genotype of this strain I-d was included in line 75. However, we have not discussed this issue further since we only used one strain, and this is not a naturally resistant strain which would be relevant to discuss. In this sense, many authors have used different strains from distinct DTUs to induce resistance (Lima et al., 2023; Mejia et al., 2008; Murta et al., 2006; Nogueira et al., 2008; Wilkinson et al., 2008), which always results in resistance strains, regardless of the parasite genotype. We believe that the relevant feature in this type of study is how the pressure is done, which will determine the different resistance mechanisms of the parasites, which were discussed in the manuscript.

● Lima, D. A., Gonçalves, L. O., Reis-Cunha, J. L., Guimarães, P. A. S., Ruiz, J. C., Liarte, D. B., & Murta, S. M. F. (2023). Transcriptomic analysis of benznidazole-resistant and susceptible Trypanosoma cruzi populations. Parasites & Vectors, 16(1), 167. https://doi.org/10.1186/s13071-023-05775-4

● Mejia, A. M., Hall, B. S., Taylor, M. C., Gómez-Palacio, A., Wilkinson, S. R., Triana-Chávez, O., & Kelly, J. M. (2012). Benznidazole-resistance in Trypanosoma cruzi is a readily acquired trait that can arise independently in a single population. The Journal of Infectious Diseases, 206(2), 220–228. https://doi.org/10.1093/infdis/jis331

● Murta, S. M. F., Krieger, M. A., Montenegro, L. R., Campos, F. F. M., Probst, C. M., Ávila, A. R., Muto, N. H., Oliveira, R. C. D., Nunes, L. R., Nirdé, P., Bruna-Romero, O., Goldenberg, S., & Romanha, A. J. (2006). Deletion of copies of the gene encoding old yellow enzyme (TcOYE), a NAD(P)H flavin oxidoreductase, associates with in vitro-induced benznidazole resistance in Trypanosoma cruzi. Molecular and Biochemical Parasitology, 146(2), 151–162. https://doi.org/10.1016/j.molbiopara.2005.12.001

● Nogueira, F. B., Krieger, M. A., Nirdé, P., Goldenberg, S., Romanha, A. J., & Murta, S. M. F. (2006). Increased expression of iron-containing superoxide dismutase-A (TcFeSOD-A) enzyme in Trypanosoma cruzi population with in vitro-induced resistance to benznidazole. Acta Tropica, 100(1–2), 119–132. https://doi.org/10.1016/j.actatropica.2006.10.004

● Wilkinson, S. R., Taylor, M. C., Horn, D., Kelly, J. M., & Cheeseman, I. (2008). A mechanism for cross-resistance to nifurtimox and benznidazole in trypanosomes. Proceedings of the National Academy of Sciences, 105(13), 5022–5027. https://doi.org/10.1073/pnas.0711014105

Have the researchers observed any variations in parasite susceptibility based on its biological stage? While epimastigotes are commonly utilized for studies, could trypomastigotes (the infective form in mammals) exhibit different responses? Additionally, have the researchers detected any differences at the transcriptomic level?

R/ Thank you for bringing this matter to our attention. In the past, we analyzed different stages in our resistance clones, and we confirmed that although the IC50 values were different between amastigotes, epimastigotes and trypomastigotes, the resistance levels were maintained in other biological stages (González, et al., 2017). In this study, we did not evaluate other forms.

● González, L., García‐Huertas, P., Triana‐Chávez, O., García, G. A., Murta, S. M. F. & Mejía‐Jaramillo, A. M. (2017). Aldo‐keto reductase and alcohol dehydrogenase contribute to benznidazole natural resistance in Trypanosoma cruzi. Molecular Microbiology, 106(5), 704–718. https://doi.org/10.1111/mmi.13830

Line 88. Please clarify what you mean by "same experimental and environmental conditions."

R/ It means that they were under the same conditions (same temperature, same medium, same number of plate-wells) as parasites treated with the drugs. To avoid confusion, we modified this as: “Untreated parasites were used as growth controls.” (Line 84)

Line 198. It would enhance the discussion by providing more information about the strains and clones analyzed.

R/ We only used one strain; from this strain, we selected one clone corresponding to the sensitive clone. After the pressure of this clone with Bz, we obtained a resistant phenotype from which we achieved 14 different clones and selected one to perform the experiments. In our results, we described all the experiments done with these two clones (one sensitive and one resistant).

Line 462. Please replace "Trypanosoma cruzi" with "T. cruzi."

R/ We replaced it.

Figures: Please enhance the quality of the figures.

R/ All the figures were exported as .tiff with a resolution of 600 dpi

Reviewers' comments:

Reviewer #1: The article presents a very novel work. It is approached from a transcriptomic approach to elucidate the possible mechanisms of resistance to BZ in populations of T. cruzi resistant by pressure to induced BZ combined with CRISPR Cas technology for genetic knock out (It was lacking to exploit these results a little more in the discussion). However, the discussion of the results needs to be strengthened in general by comparing with more related literature. Although one of the most relevant conclusions of the article is to reaffirm what has been reported by other authors, NTR I independent and multifactorial resistance mechanisms. It remains to further nurture this discussion by trying to present the possible alternative mechanisms that were rightly mentioned in the introduction (lines 44-50).

>What is the reason why the authors performed the work with BZ and not with NFX or both (not bad, but to know their reasons would be good). Why were IC50 determinations made for both, but only selective pressure against BZ? What is the reason for making this decision?

R/ We thank the reviewer for this comment. We only chose Bz to induce resistance since we had previous works with natural and induced resistance with this drug, so we thought it would be interesting to compare the results obtained here with the previous one. Also, we were interested in comparing our results with those obtained by Lima et al. (2023). Additionally, there is more published information about the treatment with Bz than with Nfx in patients as it is the only treatment available in many countries, so we believe it was more appropriate to analyze this drug. However, we confirmed that our resistance clone had cross-resistance with Nfx, which means that some of the mechanisms are true for both drugs. We think it will be engaging in the future to also use Nfx.

In the methodology it is mentioned that 10 concentrations of Bz and NFX were used, specify which were those concentrations. It also does not specify the presentation in which NFX and BZ were used, pure, tablet or purified from a tablet?

R/ Thank you for your feedback. We included this information in the manuscript as follows: “In each assay, 10 concentrations of the drugs (0.78-400 �M for Bz or 0.195-100 �M for Nfx) purified from tablets and prepared in DMSO, were evaluated. Each concentration was assessed four times in 96-well plates.” (Line 82).

>What is the justification for performing the determination of the IC50 at 72 h?

R/ The IC50 (half-maximal inhibitory concentration) is evaluated at 72 hours because it allows cells to divide 1–2 times and observe drug effects without reaching confluence and it will depend mainly on the parasite stage evaluated. In case of no replicative forms it is common to evaluate the IC50 at 24 hours.

>Specify in the methodology what was the concentration of rezarzurin used since only 20 ul/well is mentioned, but this is very ambiguous.

R/ Thank you for your feedback. We changed this in the manuscript as follows: “Plates were incubated for 72 h at 28 °C, and the effect of the compounds was determined by adding 20 �l/well of alamarBlue (Resazurin sodium salt, catalog number R7017, Sigma Aldrich, St. Louis, USA), prepared in PBS at 0.125 mg/mL and incubating overnight at 28 °C, as was described previously [19].” (Lines 84-87).

>How do they justify the use of the Epimastigote form and not the trypomastigote form?

R/ In our experience, we have noticed that the most significant changes related to this type of induced resistance are fixed in the DNA as deletions, mutations, and amplifications, which will change the gene expression in these parasites. In both stages, it is possible to find genes that modify the expression due to resistance. However, since epimastigotes are the replicative form, it will allow us to see more relevant genes involved in this and other process and also it was possible to compared our results with the ones obtained by Lima et al. (2023).

>While the focus of the article is not to relate this to intra and inter DTUs genetic variability of T. cruzi (suggested to mention this in the introduction as a possible factor that may influence heterogeneity of response to Bz), how scalable or representable are these results for the other DTU's? Would one expect a big change using another DTU or another TcI isolate?

R/ As mentioned before, despite many efforts made by our research team and other researchers, we have not found a clear correlation between drug resistance and the DTUs. We have analyzed the IC50 to Bz of many strains from different DTUs, and there is no clear pattern between these two features. Furthermore, many authors have used different strains from distinct DTUs to induce resistance, which always results in resistance strains independent of the genotype of the parasites. We believe the relevant feature in this type of study is how the pressure is done, which determines the different resistance mechanisms of the parasites, which was discussed in the manuscript. Please, find below some references that showed the mentioned above.

● Gómez-Palacio, A., Lopera, J., Rojas, W., Bedoya, G., Cantillo-Barraza, O., Marín-Suarez, J., Triana-Chávez, O. & Mejía-Jaramillo, A. (2016). Multilocus analysis indicates that Trypanosoma cruzi I genetic substructure associated with sylvatic and domestic cycles is not an attribute conserved throughout Colombia. Infection, Genetics and Evolution, 38, 35–43. https://doi.org/10.1016/j.meegid.2015.11.026

● Lima, D. A., Gonçalves, L. O., Reis-Cunha, J. L., Guimarães, P. A. S., Ruiz, J. C., Liarte, D. B., & Murta, S. M. F. (2023). Transcriptomic analysis of benznidazole-resistant and susceptible Trypanosoma cruzi populations. Parasites & Vectors, 16(1), 167. https://doi.org/10.1186/s13071-023-05775-4

● Mejia, A. M., Hall, B. S., Taylor, M. C., Gómez-Palacio, A., Wilkinson, S. R., Triana-Chávez, O., & Kelly, J. M. (2012). Benznidazole-resistance in Trypanosoma cruzi is a readily acquired trait that can arise independently in a single population. The Journal of Infectious Diseases, 206(2), 220–228. https://doi.org/10.1093/infdis/jis331

● Mejia-Jaramillo, A. M., Fernández, G. J., Montilla, M., Nicholls, R. S., Triana-Chávez, O. (2012). Sensibilidad al benzonidazol de cepas Trypanosoma cruzi sugiere la circulación de cepas naturalmente resistentes en Colombia. Biomédica, 32, 196-205.

● Murta, S. M. F., Krieger, M. A., Montenegro, L. R., Campos, F. F. M., Probst, C. M., Ávila, A. R., Muto, N. H., Oliveira, R. C. D., Nunes, L. R., Nirdé, P., Bruna-Romero, O., Goldenberg, S., & Romanha, A. J. (2006). Deletion of copies of the gene encoding old yellow enzyme (TcOYE), a NAD(P)H flavin oxidoreductase, associates with in vitro-induced benznidazole resistance in Trypanosoma cruzi. Molecular and Biochemical Parasitology, 146(2), 151–162. https://doi.org/10.1016/j.molbiopara.2005.12.001

● Nogueira, F. B., Krieger, M. A., Nirdé, P., Goldenberg, S., Romanha, A. J., & Murta, S. M. F. (2006). Increased expression of iron-containing superoxide dismutase-A (TcFeSOD-A) enzyme in Trypanosoma cruzi populatio

---

## [Decision Letter · Decision Letter 1]

19 Jan 2025

PONE-D-24-50847R1

Transcriptomic analysis of benznidazole-resistant Trypanosoma cruzi clone reveals nitroreductase I-independent resistance mechanismsPLOS ONE

Dear Dr. Ana Maria Mejia-Jaramillo,

Thank you for submitting your manuscript to PLOS ONE. After careful consideration, we feel that it has merit but does not fully meet PLOS ONE’s publication criteria as it currently stands. Therefore, we invite you to submit a revised version of the manuscript that addresses the points raised during the review process.

We look forward to receiving your revised manuscript.

Kind regards,

Claudia Patricia Herrera, Ph.D.

Academic Editor

PLOS ONE

Journal Requirements:

Additional Editor Comments:

Thank you for submitting the revision and addressing the reviewers' comments. I believe the paper is ready for publication; however, I would appreciate it if you could take the last comment from Reviewer 2 into account before final acceptance.

Reviewers' comments:

Reviewer's Responses to Questions

**Comments to the Author**

1. If the authors have adequately addressed your comments raised in a previous round of review and you feel that this manuscript is now acceptable for publication, you may indicate that here to bypass the “Comments to the Author” section, enter your conflict of interest statement in the “Confidential to Editor” section, and submit your "Accept" recommendation.

Reviewer #1: (No Response)

Reviewer #2: All comments have been addressed

2. Is the manuscript technically sound, and do the data support the conclusions?

Reviewer #1: Yes

Reviewer #2: Yes

3. Has the statistical analysis been performed appropriately and rigorously? 

Reviewer #1: Yes

Reviewer #2: Yes

4. Have the authors made all data underlying the findings in their manuscript fully available?

Reviewer #1: Yes

Reviewer #2: Yes

5. Is the manuscript presented in an intelligible fashion and written in standard English?

Reviewer #1: Yes

Reviewer #2: Yes

6. Review Comments to the Author

Reviewer #1: I have read the revised version, and it is much better and more scientifically robust, I would just like to point out two things to the authors.

1. in relation to resarsurin, the authors state the use of 20 uL of the above mentioned reagent but they still omit the concentration they have of the compound in those 20 uL, please specify.

2. Has recently been published the work; Ochoa-Martínez P, López-Monteon A, López-Domínguez J, Manning-Cela RG, Ramos-Ligonio A. Expression Analysis of Thirteen Genes in Response to Nifurtimox and Benznidazole in Mexican Isolates of Trypanosoma cruzi by Digital PCR. Acta Parasitol. 2025 Jan 7;70(1):15. doi: 10.1007/s11686-024-00986-w. Where a review of the activation of various genes in response to stress exerted by BZN and NFX is made, what can the authors discuss their results in relation to this work?

Reviewer #2: The author has thoroughly addressed all the suggestions and recommendations provided. Upon further review, I am confident that this article is now suitable for publication in PLOS ONE.

7. PLOS authors have the option to publish the peer review history of their article (what does this mean? ). If published, this will include your full peer review and any attached files.

**Do you want your identity to be public for this peer review?** For information about this choice, including consent withdrawal, please see our Privacy Policy .

Reviewer #1: No

Reviewer #2: **Yes: ** Priscila Silva Grijo Farani

---

## [Author Response · Author response to Decision Letter 2]

20 Jan 2025

January 20 2025

Claudia Patricia Herrera, Ph.D.

Academic Editor

PLOS ONE

Dear Editor:

We thank the editor and reviewers for taking the time to review our manuscript PONE-D-24-50847. Their comments have undoubtedly improved the quality of our manuscript. We are glad that they found our manuscript of interest, and our responses to their comments are provided below. In the new version of the manuscript, the changes suggested by the editors and reviewers are indicated in red.

Sincerely,

Prof. Ana María Mejía Jaramillo

Universidad de Antioquia

Medellín, Colombia

General changes

Journal Requirements:

R/ The references where revised.

Additional Editor Comments:

Thank you for submitting the revision and addressing the reviewers' comments. I believe the paper is ready for publication; however, I would appreciate it if you could take the last comment from Reviewer 2 into account before final acceptance.

R/ Thank you. The two comments where addressed. You will find the answer below.

Reviewers' comments:

Reviewer #1: I have read the revised version, and it is much better and more scientifically robust, I would just like to point out two things to the authors.

1. in relation to resarsurin, the authors state the use of 20 uL of the above mentioned reagent but they still omit the concentration they have of the compound in those 20 uL, please specify.

R/ Thank you for your feedback. In the last version, we did include the concentration which is 0.125 mg/mL as follows: “Plates were incubated for 72 h at 28 °C, and the effect of the compounds was determined by adding 20 �l/well of alamarBlue (Resazurin sodium salt, catalog number R7017, Sigma Aldrich, St. Louis, USA), prepared in PBS at 0.125 mg/mL and incubating overnight at 28 °C, as was described previously [19].” (Lines 84-87).

2. Has recently been published the work; Ochoa-Martínez P, López-Monteon A, López-Domínguez J, Manning-Cela RG, Ramos-Ligonio A. Expression Analysis of Thirteen Genes in Response to Nifurtimox and Benznidazole in Mexican Isolates of Trypanosoma cruzi by Digital PCR. Acta Parasitol. 2025 Jan 7;70(1):15. doi: 10.1007/s11686-024-00986-w. Where a review of the activation of various genes in response to stress exerted by BZN and NFX is made, what can the authors discuss their results in relation to this work?

R/ Thank you for bringing this matter to our attention. We included the reference in the paper and discussed as follow: “However, a recent study using Mexican T. cruzi isolates with different Bz and NFx resistance profiles showed by digital PCR that resistant parasites had increased NTRI expression. This was counterbalanced by overexpression of detoxifying enzymes such as SOD and GTS or transporters such as MDR [44]. (Lines 507-511).

44. Ochoa-Martínez P, López-Monteon A, López-Domínguez J, Manning-Cela RG, Ramos-Ligonio A. Expression analysis of thirteen genes in response to nifurtimox and benznidazole in Mexican isolates of Trypanosoma cruzi by digital PCR. Acta Parasit. 2025;70: 1–11. doi:10.1007/s11686-024-00986-w

Reviewer #2: The author has thoroughly addressed all the suggestions and recommendations provided. Upon further review, I am confident that this article is now suitable for publication in PLOS ONE.

R/ We are very grateful for the comments on our manuscript.

---

## [Editor Report · Decision Letter 2]

26 Jan 2025

Transcriptomic analysis of benznidazole-resistant Trypanosoma cruzi clone reveals nitroreductase I-independent resistance mechanisms

PONE-D-24-50847R2

Dear Dr. Ana Maria Mejia-Jaramillo

We’re pleased to inform you that your manuscript has been judged scientifically suitable for publication and will be formally accepted for publication once it meets all outstanding technical requirements.

Kind regards,

Claudia Patricia Herrera, Ph.D.

Academic Editor

PLOS ONE
---

## [Editor Report · Acceptance letter]

PONE-D-24-50847R2

PLOS ONE

Dear Dr. Mejia-Jaramillo,

I'm pleased to inform you that your manuscript has been deemed suitable for publication in PLOS ONE. Congratulations! Your manuscript is now being handed over to our production team.

Kind regards,

on behalf of

Dr. Claudia Patricia Herrera

Academic Editor

PLOS ONE